# Performance of an offline systematic error correction strategy in pediatric patients receiving adjuvant conformal radiotherapy for Wilm's tumor

**Rashid Mussa Mruma**[1]*, **Nazima Dharsee**[1,2], **Christina Vallen Malichewe**[1], **Jumaa Dachi Kisukari**[1,2], **Furahini Yoram**[1], **Hemed Said Myanza**[2], **Stephen Shedrack Meena**[2], **Geofrey Filbert Soko**[1,2]

1 Department of Clinical Oncology, Muhimbili University of Health and Allied Sciences, Dar es Salaam, Tanzania, 2 Ocean Road Cancer Institute, Dar es Salaam, Tanzania

* semruma2000@yahoo.com

**Data Availability Statement:** All relevant data are within the manuscript and its Supporting Information files.

## Abstract

### Background

Radiotherapy plays a key role as an adjuvant treatment in pediatric Wilm's tumor, improving both survival and quality of life. The success of radiotherapy depends on the precise delivery of radiation dose to the tumor while sparing radiosensitive structures in the vicinity of the tumor. Pediatric patients pose unique challenges in achieving accurate radiotherapy delivery due to their inability to understand instructions and the high radiosensitivity of their tissues. Thus, it is important to determine the optimum geometric verification strategy that will ensure accurate delivery of the prescribed target as specified in the patient's treatment plan.

### Purpose

To evaluate the performance of an offline geometric correction strategy in ensuring accuracy and reproducibility during radiotherapy delivery in Wilm's tumor patients.

### Material and methods

The extended no-action level offline correction strategy was applied in the radiotherapy delivery of 45 Wilm's tumor patients. Gross errors from the first three fractions were used to calculate the mean errors which were then applied as offline correction factors. Mean errors among different groups were compared using a two-way analysis of variance (ANOVA) and Dunnett's pairwise comparisons. All statistical analyses and data visualization were performed using GraphPad Prism version 7 (Insight Partners, GraphPad Holdings, LLC).

### Results

A total of 45 patients were included in the study. In all three orthogonal directions, the recorded gross errors were significantly lower after the application of the systematic error corrections. Random errors were significantly larger in the longitudinal direction compared

**Funding:** The author(s) received no specific funding for this work.

**Competing interests:** The authors have declared that no competing interests exist.

to lateral (mean difference = 0.28, *p = 0.036*) and vertical directions (mean difference = 0.37 cm, p = 0.003). Patients' age was a significant predictor of random errors whereby the magnitude of random error decreased with increasing age.

## Conclusion

This study shows that the offline correction strategy used is effective in ensuring the accuracy of radiotherapy delivery in pediatric Wilm's tumor patients.

## Introduction

Wilm's tumor is a common malignancy in the pediatric population that is associated with high mortality rates [1]. Global disparities exist in Wilm's tumor incidence and treatment outcomes whereby low and middle-income countries are experiencing higher incidences and mortality compared to high-income countries [2, 3]. The number of children diagnosed with Wilm's tumor in Tanzania has been increasing in recent years [4]. Therefore, concerted efforts are needed to improve early detection through screening as well as ensure the availability of optimum treatment.

Adjuvant radiotherapy following surgical excision of the affected kidney is associated with significant improvements in both survivals as well as the quality of life in children with high-risk Wilm's tumors [5]. Despite the well-documented beneficial effects of ionizing radiation in killing cancer cells which can lead to cure or palliation of symptoms, radiation is equally destructive to normal tissues that are close to the tumor. Therefore the clinical success of radiotherapy is based on the ability to deliver a tumoricidal radiation dose while ensuring a minimum dose to the normal tissues surrounding the tumor volume [6]. To achieve this premise, careful treatment planning should be done to ensure well-optimized dose distributions to the target and surrounding organs at risk. Radiotherapy planning has undergone major improvements in recent years with the introduction of conformal radiotherapy techniques such as Three-Dimensional Conformal Radiotherapy (3D CRT), Intensity Modulated Radiotherapy (IMRT), Volumetric Modulated Arc Therapy (VMART) and Stereotactic Body Radiotherapy (SBRT) which have better dose distributions than traditional two-field flank irradiation techniques [7, 8]. Equally important is ensuring the planned treatment is delivered to the patient with minimal uncertainties which require reliable immobilization as well as geometrical and dosimetric treatment verification.

Imaging plays a key role in ensuring the geometrical accuracy of radiotherapy delivery. Imaging modalities including Megavoltage (MV) Electronic Portal Imaging Devices (EPID), onboard x-ray imagers, Cone beam CT, ultrasound, and Magnetic Resonance Imaging (MRI) provide images of the patient in treatment position that can be matched with reference images generated during planning to correct errors in daily positioning during treatment [9]. This kind of verification can be done using an online strategy whereby imaging and error corrections are performed during each treatment session or offline whereby imaging and error correction are performed in pre-determined treatment fractions. Online correction strategies are the most effective in ensuring accuracy in radiotherapy delivery but their use may be constrained by practical issues such as the type of available imaging modality, the number of patients treated per treatment machine as well as resource availability [10]. Therefore, well-established and tested offline correction strategies are needed in this situation to ensure radiation is delivered accurately even during fractions where imaging is not done.

One common offline correction protocol is the extended no-action level protocol [11] originally introduced by de Boer and Heijmen which requires imaging on the first n treatment fractions (where n is usually $\geq$ 3) followed by determination of the systematic error and applying a correction for the remainder of the treatment fractions. Further imaging is done once weekly to confirm that gross errors are within tolerance levels set out by departmental protocols. This protocol has shown acceptable performance in the majority of treatment sites in adult patients [12, 13]. Despite the well-documented good performance of this protocol in adult patients, data on its performance in pediatric patients are lacking.

Pediatric patients in particular pose unique challenges due to their inability to understand and follow instructions and the high radiosensitivity of their tissues. Thus, it is important to determine the optimum geometric verification strategy that will ensure accurate delivery of the prescribed target as specified in the patient's treatment plan. Therefore, here we report the performance of the extended no-action level protocol when applied in pediatric patients with Wilm's tumor.

# Methods and materials

## Study design and setting

A retrospective study was conducted to evaluate the performance of offline geometric correction strategy in Wilm's tumor patients treated at Ocean Road Cancer Institute (ORCI) from January to December 2021. Normally Wilm's tumor patients undergo 3D CRT of the whole abdomen with a total dose of 18 Gy in 12 fractions.

## Study participants

This study included 45 Wilm's tumor patients who were referred to ORCI for adjuvant radiotherapy treatment from January to December 2021. All Wilm's tumor patients in this study received adjuvant radiotherapy after undergoing a nephrectomy to remove the affected kidney. All patients were treated in a vital beam linear accelerator using 6 MV photons (Varian Medical Systems, Palo Alto, United States).

## Pre-treatment preparation

All patients underwent CT simulation while immobilized in a thermoplastic shell covering their abdomen and pelvic areas. Radiopaque markers were placed at laser intersection points in the anterior and lateral aspects of the patients on the thermoplastic shells and they were then used to define the user origin during the first treatment fraction. Scanning was done using 3 mm slices from the lower thorax to the upper pelvis.

CT images of the patients were then transferred to the Eclipse treatment planning system (Varian Medical Systems, Palo Alto, United States) for the definition of target and organ at risk volumes as well as dosimetry planning. All Wilm's tumor patients in this study received post-surgery radiotherapy after undergoing a nephrectomy to remove the affected kidney and therefore no Gross Tumor Volume (GTV) was identified. The Clinical Tumor Volume (CTV) corresponded to the entire abdominal cavity from the dome of the diaphragm to the pelvic floor, then a 10 mm margin was applied to the CTV to define the Planning Tumor Volume (PTV) following recommendations of the International Society of Paediatric Oncology's Renal Tumour Study Group (SIOP-RTSG) [8]. The contralateral kidney, spinal cord and liver were contoured as organs at risk following the Radiation Oncology Group (RTOG) guidelines [14]. Conformal radiotherapy plans were then created with the prescribed dose of 18 Gy in 12 fractions of 1.5 Gy [15].

Three-dimensional conformal radiotherapy (3DCRT) plans were created using opposing antero-posterior beams. Plans were optimized by using MLCs to shield organs at risk including the contralateral kidney and the liver. Additional fields as well as in-fields were added to increase dose coverage to specific areas of the PTV. Dose-volume histograms (DVHs) and dose colour washes were generated to evaluate the 3D dose distributions and the acceptable PTV dose heterogeneity was 95 to 107%.

## Radiotherapy delivery and treatment verification

Patients were positioned and immobilized on the treatment couch following the routine departmental protocol. During the first treatment fraction, patients were first aligned using alignment marks at CT simulation on their thermoplastic shells. From these reference marks, couch shifts were manually applied in three directions (longitudinal, lateral, and vertical) directions to obtain the treatment iso-centre. This isocenter was marked, and then MV images were taken using the EPID and immediately matched with the Digitally Reconstructed Radiography (DRR) using image viewing and matching features of the treatment machine console. Image matching was confirmed by checking the superimposition of the vertebral column, obturator foramina and iliac crests. Gross errors were then corrected and followed by the delivery of the first treatment fraction. During the second and third treatment fractions, patients were positioned using the isocenter marks created during the first treatment session followed by imaging and image matching, gross error correction, and treatment delivery. After the third treatment session, the systematic error was calculated in the three orthogonal directions. The calculated systematic errors were then applied as correction factors for subsequent treatment fractions. For the rest of treatment fractions, daily imaging was performed but corrections were only applied when recorded gross errors exceeded the action level.

## Systematic error correction strategy

The extended no-action level correction strategy was applied to the fourth treatment fraction. In this strategy, gross errors from a given number of treated fractions are used to calculate the mean of the gross errors (correction factor) in three directions. The correction factor was calculated as the average of gross errors recorded during the first three fractions as given by Eq (i).

$$\bar{X} = \frac{\sum X}{n} \tag{i}$$

Where $\bar{X}$ = individual patient systematic error (correction factor)
$\sum$ = summation
n = number of observations = 3
x = gross error

The mean of the gross errors is then applied as correction factors for the remainder of the treatment fractions. To apply this strategy, gross errors were retrieved from the record management system and were used to calculate systematic errors. Patients were positioned using alignment markers created during the first treatment fraction followed by the application of correction factors equal in magnitude and direction to the calculated systematic errors. The new corrected treatment iso-centers were re-marked and this new information was saved permanently in the Aria record management system to be used for the remaining treatment fractions. Further verification imaging was done once weekly to confirm compliance with tolerance margins.

### Data collection and analysis

For every patient, gross errors were retrieved from the Aria record management system between November 2022 and January 2023. Then these errors were used for further statistical analysis. Data was first entered into a spreadsheet which was then imported to GraphPad Prism for data visualization and statistical analysis. Two-way analysis of variance (ANOVA) and Dunnett's pairwise comparisons were used to test the statistical significance of the relationship between calculated errors and patient characteristics in the three orthogonal directions. All data were analysed using GraphPad Prism version 7 (Insight Partners, GraphPad Holdings, LLC).

### Ethical approval

Ethics approval was obtained from the Muhimbili University of Health and Allied Sciences institutional research ethics review board and permission was granted by the Ocean Road Cancer Institute to access patient data and use them for this study.

## Results

### Socio-demographic characteristics of study subjects

A total of 45 patients were included in this study, the majority (73.5%) were aged 1 to 5 years. The majority (75.6%) of the included patients were in stage III of the disease, while curative indication was the most common accounting for 93.3% (Table 1).

### Distribution of gross errors

Gross errors were recorded in the longitudinal, lateral, and vertical directions in all included patients after superimposition of the MV and DRR images (Fig 1A and 1B). Gross errors were recorded in four separate treatment fractions (the first three fractions and the sixth fraction). Recorded gross errors were larger in the longitudinal (Fig 1C) and lateral directions (Fig 1D) compared to the vertical direction (Fig 1E). In all three orthogonal directions, the recorded gross errors were significantly lower after the application of the systematic error corrections and were within the tolerance of departmental protocol for gross errors in pediatric Wilm's tumour patients (Fig 1C–1E).

**Table 1. Socio-demographic characteristics and clinical information (N = 45).**

| Variable | Categories | Frequency | Percentage |
|---|---|---|---|
| **Age** | 1–5 | 33 | 73.5 |
| | 6–10 | 11 | 24.3 |
| | 11–15 | 1 | 2.2 |
| **Sex** | Male | 20 | 44.4 |
| | Female | 25 | 55.6 |
| **Indication** | Curative | 42 | 93.3 |
| | Palliative | 3 | 6.7 |
| **Stage of the disease** | I | 1 | 2.2 |
| | II | 4 | 8.9 |
| | III | 34 | 75.6 |
| | IV | 6 | 13.3 |

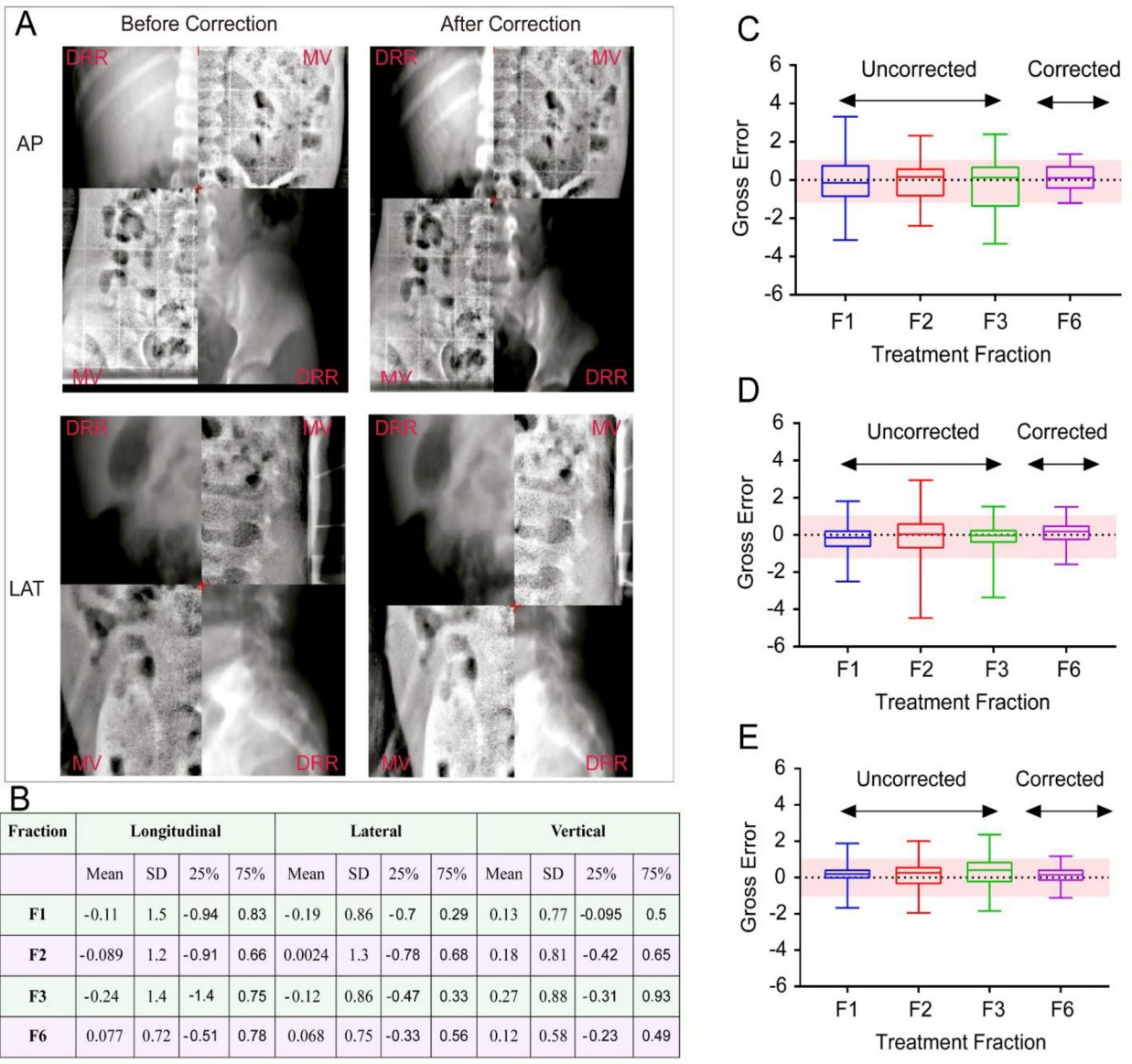

**Fig 1. Gross error determination according to employed off-line correction protocol.** A, Representative Split-window view of blended images (DRR and MV as labelled on image sections) before and after correction of gross error for one patient in one fraction. The displacements of the MV-images relative to the DRR were recorded in all three orthogonal directions (longitudinal, lateral, and vertical) and their values constituted gross errors. B, Summary statistics of the gross errors in respective treatment fractions (F1, F2, F3 & F6) including Mean, Standard Deviation, and 25 & 75 percentiles. C-E show the distribution of gross errors over the first three treatment fractions (F1-F3) and the sixth fraction (F6) in the three orthogonal directions (C-longitudinal, D-lateral, and E-vertical). The shaded region represents the tolerance of departmental protocol.

### Calculated random and systematic errors

Random errors were significantly larger in the longitudinal direction compared to lateral (mean difference = 0.28, $p = 0.036$) and vertical directions (mean difference = 0.37 cm, p = 0.003). The number of extreme outliers (>2 cm) was also higher in the longitudinal direction compared to lateral and vertical directions (Fig 2A). Furthermore, systematic errors were higher in magnitude in the vertical direction (mean = 0.19) compared to longitudinal (mean = -0.14) and lateral (mean = -0.1) directions with the lateral direction having the lowest magnitude (Fig 2B). However, the differences were not statistically significant. The number of

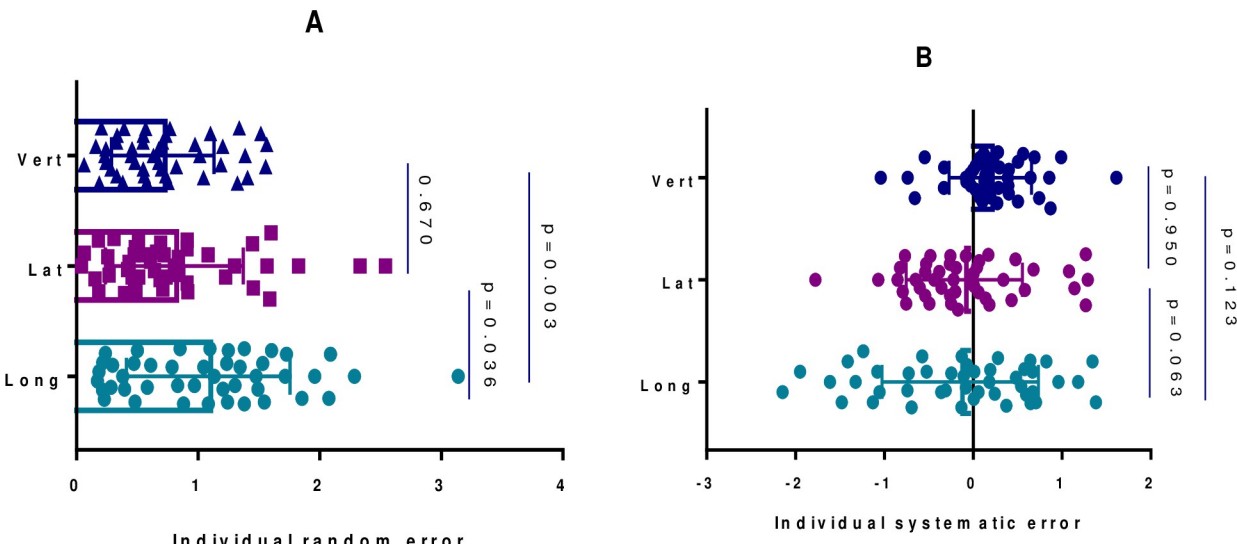

**Fig 2. Calculated random and systematic errors in the three orthogonal directions (longitudinal, lateral, and vertical).** A, Random errors which were calculated as the standard deviation of the gross errors recorded in the first three treatment fractions for every individual patient. B, Systematic errors which were calculated by taking the average of the gross errors recorded in the first three treatment fractions for every individual patient. The calculated mean of the gross errors was then used as correction factors for the remainder of the treatment fractions.

extreme outliers ($>1$ cm) was higher in the longitudinal direction compared to the other two directions.

## Determinants of random and systematic errors

Results of this study show that a patient's age is a significant predictor of random errors whereby the magnitude of random error decreases with increasing age (Fig 3D). Other factors including sex and disease stage did not significantly correlate with systematic or random errors (Fig 3).

## Discussion

The therapeutic ratio of radiotherapy depends on two main variables that include the dose delivered to the tumor which determines the cure probability and the dose delivered to the normal tissues which determines the probability of treatment complications [16]. Therefore, the precise delivery of radiation dose to the tumour while sparing normal tissues is of paramount importance in clinical radiotherapy. In this study, the performance of the offline extended no action level correction protocol [11] in ensuring accuracy and precision in the delivery of adjuvant radiotherapy for pediatric Wilm's tumour patients was evaluated.

We first evaluated the gross errors in the first three days and used them to calculate the systematic errors that were then applied as correction factors for the upcoming fractions. We then compared the gross errors in the first three fractions to those of the sixth fraction in all three orthogonal directions. The results of this study show a significant reduction of gross errors in the sixth fraction compared to the first three fractions in all three orthogonal directions. The mean, standard deviation as well as the 25th and 75th percentiles were significantly lower in the sixth fraction compared to the first three fractions and the gross errors always lay within the tolerance value set out by our departmental protocol. These results prove that the extended no-action level protocol is an effective correction strategy in this patient cohort and are in agreement with the results of similar studies in other patient cohorts [12, 13].

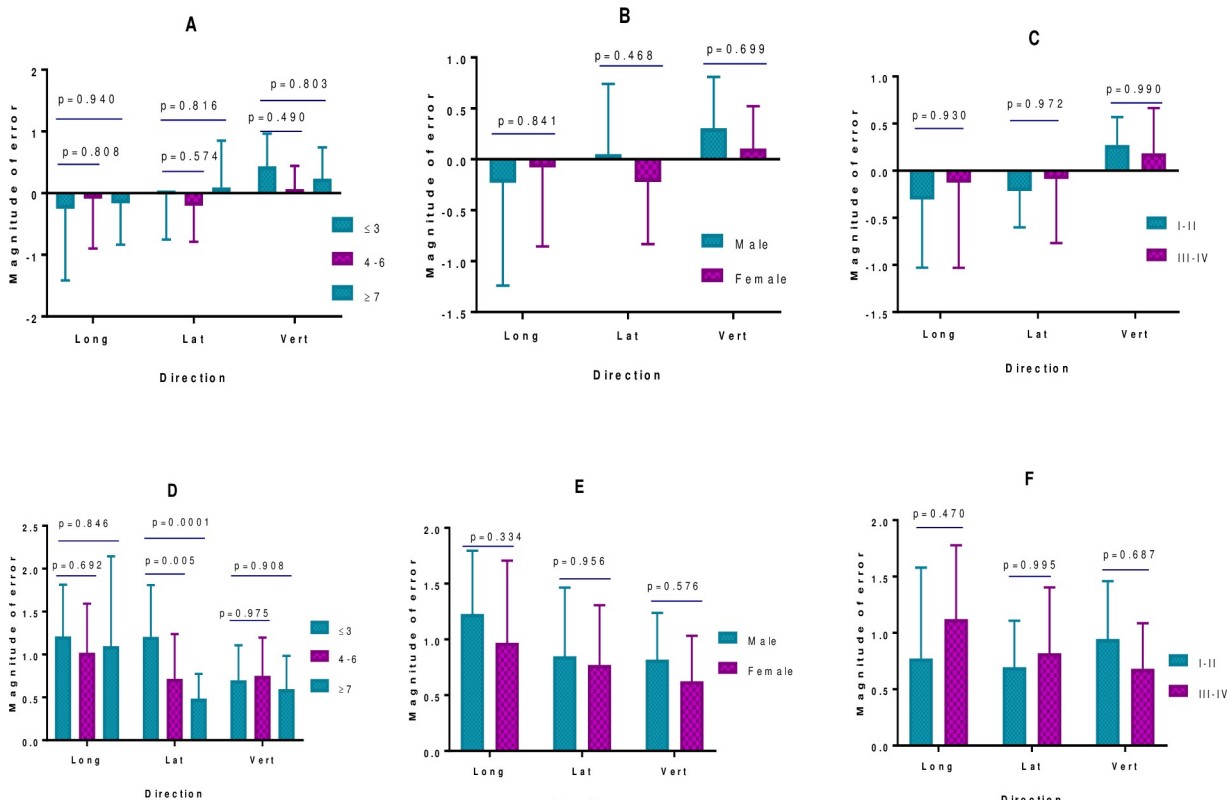

**Fig 3. Determinants of calculated systematic and random errors.** Calculated systematic and random errors were compared by age, sex, and stage of disease using two-way analysis of variance (ANOVA) and Dunnett's pairwise comparisons. A-C show systematic errors by age, sex, and stage respectively. D-F show random errors by age, sex, and stage respectively. Error bars represent the standard deviation of calculated errors in each group.

We also calculated the random and systematic components of the recorded gross errors and applied the mean from the gross errors as correction factors for the remaining treatment fractions. We found similar contributions from random and systematic errors in the overall gross errors. Random errors were larger in the longitudinal direction compared to the other two orthogonal directions. Furthermore, systematic errors did not differ significantly across the three orthogonal directions. However, in other studies employing similar correction protocols, significant differences were found in the relative contributions of systematic and random components in the overall gross error [17, 18]. In the study by Eldebawy et al. (2011), the random component was larger than the systematic component while the opposite was true in the study by Molana et al. (2018). These findings imply that they offer information on which components of the treatment procedure need to be scrutinized to reduce the observed errors. Furthermore, the overall impact of random and systematic errors on delivered doses are not the same whereby systematic errors are more detrimental. To improve accuracy in radiotherapy delivery, proper patient positioning equipment, immobilization techniques, increasing laser alignment, and correcting couch and gantry stability are essential [19].

Results of this study showed that patients' age is the significant predictor of random errors whereby errors were larger in younger patients compared to older patients. This implies that although the protocol showed overall good performance in the study population, special considerations should be taken with very young patients ($\leq$ 3 years). These considerations may include daily imaging in the form of online correction or more robust immobilization or sedation to ensure limited intrafraction as well as interfraction errors [20].

## Conclusion

The results of this study show that the use of the extended no-action level protocol (eNAL) is effective in correcting systematic errors and ensuring radiotherapy is delivered within the uncertainty margins set out by the departmental protocol for pediatric Wilm's tumor patients receiving whole abdominal radiotherapy. This protocol can be used for a range of ages but for younger children alternative verification strategies like the use of online verification, sedation, and surface-guided radiotherapy should be considered to limit movement.

## Supporting information

**S1 Raw data.**
(XLSX)

## Acknowledgments

The authors would like to acknowledge all Radiotherapists at Ocean Road Cancer Institute who were working in the linear accelerator machines during data collection for their contribution to treatment verification and data collection.

## Author Contributions

**Conceptualization:** Rashid Mussa Mruma, Nazima Dharsee, Christina Vallen Malichewe, Jumaa Dachi Kisukari, Furahini Yoram, Hemed Said Myanza, Geofrey Filbert Soko.

**Data curation:** Rashid Mussa Mruma, Nazima Dharsee, Jumaa Dachi Kisukari, Hemed Said Myanza, Geofrey Filbert Soko.

**Formal analysis:** Rashid Mussa Mruma, Furahini Yoram, Stephen Shedrack Meena, Geofrey Filbert Soko.

**Methodology:** Rashid Mussa Mruma, Nazima Dharsee, Christina Vallen Malichewe, Furahini Yoram, Hemed Said Myanza, Stephen Shedrack Meena, Geofrey Filbert Soko.

**Project administration:** Nazima Dharsee.

**Resources:** Rashid Mussa Mruma.

**Software:** Rashid Mussa Mruma, Stephen Shedrack Meena, Geofrey Filbert Soko.

**Supervision:** Nazima Dharsee, Christina Vallen Malichewe, Jumaa Dachi Kisukari, Geofrey Filbert Soko.

**Validation:** Furahini Yoram.

**Writing – original draft:** Rashid Mussa Mruma, Geofrey Filbert Soko.

**Writing – review & editing:** Rashid Mussa Mruma, Nazima Dharsee, Christina Vallen Malichewe, Jumaa Dachi Kisukari, Furahini Yoram, Hemed Said Myanza, Stephen Shedrack Meena, Geofrey Filbert Soko.

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
