## [Decision Letter · Decision Letter 0]

23 Oct 2023

PONE-D-23-19186Performance of an Offline Systematic Error Correction Strategy in Pediatric Patients Receiving Adjuvant Conformal Radiotherapy for Wilm’s TumorPLOS ONE

Dear Dr. Mruma,

Thank you for submitting your manuscript to PLOS ONE. After careful consideration, we feel that it has merit but does not fully meet PLOS ONE’s publication criteria as it currently stands. Therefore, we invite you to submit a revised version of the manuscript that addresses the points raised during the review process.

We look forward to receiving your revised manuscript.

Kind regards,

Bruno Fionda

Academic Editor

PLOS ONE

Journal Requirements:

Reviewers' comments:

Reviewer's Responses to Questions

**Comments to the Author**

1. Is the manuscript technically sound, and do the data support the conclusions?

Reviewer #1: Yes

Reviewer #2: Yes

2. Has the statistical analysis been performed appropriately and rigorously? 

Reviewer #1: Yes

Reviewer #2: Yes

3. Have the authors made all data underlying the findings in their manuscript fully available?

Reviewer #1: Yes

Reviewer #2: Yes

4. Is the manuscript presented in an intelligible fashion and written in standard English?

Reviewer #1: Yes

Reviewer #2: Yes

5. Review Comments to the Author

Reviewer #1: The authors report their technique to minimize radiation scatter injury to surrounding tissues in the treatment of pediatric Wilms tumor. They determine gross error in their first three dose fractions and determine the gross error on each of the sessions and develop a systematic error correction factor to be applied to subsequent doses. They do so by using a combination MV EPID (Mega Voltage-Electronic Portal Imaging Devices) image acquisition and DRR (Digital reconstruction) overlay.

The paper is generally well written, and the results hold value. My questions for the authors are:

1). In Wilms tumor the right and left side have different organs at risk during radiation planning. The authors have not reported any difference between sides. One would imagine that the liver will be a more significant factor in right sided Wilms as compared to left sided tumors.

2). Why did the authors not test this further out in the dose fractions after the 6th fraction. How are they sure that there will be no further errors developing in the 9th or 12th fraction? This is particularly important as this has not been reported in children previously.

3). The authors report that longitudinal measurements had the greatest error measurements. Is this simply because this is the greatest measurement? Can the authors give their reasons as to why longitudinal measurements had greater error rates as compared to vertical and lateral measurements? Authors should also let readers know what the bony landmarks were for these respective measurements.

4). The authors measure a 0.5mm margin to their clinical tumor volume in determining their planning tumor volume. This seems a very small margin. Does this small number increase the gross error estimation?

Reviewer #2: This manuscript the effectiveness of a systematic error correction approach used for pediatric patients undergoing adjuvant conformal radiotherapy to treat Wilm's tumour. The title accurately reflects the study's content, which focuses on the unique challenges of treating children and how to address them. The conclusion is sound. Overall, I believe the paper meets standards, though some minor revisions could enhance its clarity.

The study mentions that the patients are immobilised using thermoplastic masks. Has the author considered or assessed the use of vacuum bags for immobilising these patients and if they believe it would improve setup accuracy? Additionally, surface-guided radiotherapy could be a viable option to enhance setup and treatment accuracy for these patients. It would be helpful to include information regarding this in the study.

My detailed questions are listed below:

It is unclear why daily imaging isn't being performed to improve treatment accuracy using kV orthogonal images. However, it could be due to the technology available, please add more details about it.

Are patients being imaged with the MV beam, and are imaging MUs accounted for in planned doses? If images are acquired using the treatment field, why aren't daily images considered? Is the total setup error applied when images are taken?

It would be helpful to have more details about treatment planning, including the number of fields used and field arrangement.

6. PLOS authors have the option to publish the peer review history of their article (what does this mean?). If published, this will include your full peer review and any attached files.

Reviewer #1: **Yes: **Sathyaprasad Burjonrappa

Reviewer #2: No

---

## [Author Response · Author response to Decision Letter 0]

17 Nov 2023

Reviewer #1: The authors report their technique to minimize radiation scatter injury to surrounding tissues in the treatment of pediatric Wilms tumor. They determine gross error in their first three dose fractions and determine the gross error on each of the sessions and develop a systematic error correction factor to be applied to subsequent doses. They do so by using a combination of MV EPID (Mega Voltage-Electronic Portal Imaging Devices) image acquisition and DRR (Digital reconstruction) overlay.

The paper is generally well written, and the results hold value. My questions for the authors are:

1). In Wilms tumor the right and left side have different organs at risk during radiation planning. The authors have not reported any difference between the sides. One would imagine that the liver will be a more significant factor in right-sided Wilms as compared to left-sided tumors.

Response: We acknowledge the reviewer's comment regarding the different organs at risk (OARs) on the right and left sides during planning. However, it's worth noting that the planning target volume (PTV) for all patients in this study consisted of the entire abdomen, which means that the organs at risk remained the same regardless of the side of nephrectomy. Additionally, we ensured that all treatment plans met the constraints outlined in our institutional protocol during the planning process. We have added the details of the treatment planning and optimization process in the methods and materials section

2). Why did the authors not test this further out in the dose fractions after the 6th fraction? How are they sure that there will be no further errors developing in the 9th or 12th fraction? This is particularly important as this has not been reported in children previously.

Response: Our study proposes an effective offline correction strategy for radiotherapy delivery. The strategy involves imaging on a predetermined number of days, followed by systematic error correction. This approach eliminates the need for imaging in every treatment fraction, leading to more efficient radiotherapy delivery. In cases where imaging is performed in every treatment fraction, our strategy provides reliable set-up points that reduce the time required for daily set-up corrections. We imaged all 12 fractions, but in this report, we present results from a selected number of treatment fractions.

3). The authors report that longitudinal measurements had the greatest error measurements. Is this simply because this is the greatest measurement? Can the authors give their reasons as to why longitudinal measurements had greater error rates as compared to vertical and lateral measurements? Authors should also let readers know what the bony landmarks were for these respective measurements.

Response: The larger systematic errors in the longitudinal and lateral directions, as compared to the vertical direction, are primarily due to the immobilization device used and the geometrical characteristics of the treatment machine. The vertical direction is the most stable of the three dimensions, owing to the ease of patient alignment, dependence on couch height, and increased stability of the thermoplastic mask around the anterior isocentric point. We have also included the bony landmarks used for image matching in the Radiotherapy delivery and treatment verification section.

4). The authors measure a 0.5mm margin to their clinical tumor volume in determining their planning tumor volume. This seems a very small margin. Does this small number increase the gross error estimation?

Response: We thank the reviewer for this observation. The quoted margin of 0.5 mm was due to an error, we have corrected this to the true expansion margin from CTV to PTV of 10 mm. This margin does not affect the magnitude or direction of delivery errors. However, the errors during delivery can indicate whether the margin between CTV and PTV is sufficient, and the action level should always remain below this margin.

Reviewer #2: This manuscript the effectiveness of a systematic error correction approach used for pediatric patients undergoing adjuvant conformal radiotherapy to treat Wilm's tumor. The title accurately reflects the study's content, which focuses on the unique challenges of treating children and how to address them. The conclusion is sound. Overall, I believe the paper meets standards, though some minor revisions could enhance its clarity.

The study mentions that the patients are immobilised using thermoplastic masks. Has the author considered or assessed the use of vacuum bags for immobilising these patients and if they believe it would improve setup accuracy?

Response: The reviewer is right that all patients included in this study were immobilized using a thermoplastic mask. We have previously used vacuum bags for immobilizing this group of patients. Our experience showed that vacuum bags are liable to greater rotational and translational uncertainties. Therefore, we immobilize all of our Wilm’s tumor patients using thermoplastic masks. 

Additionally, surface-guided radiotherapy could be a viable option to enhance setup and treatment accuracy for these patients. It would be helpful to include information regarding this in the study.

We agree with the reviewer's suggestion on the potential of surface-guided radiotherapy (SGRT) to enhance accuracy in our patients. Unfortunately, this is currently not possible in our settings due to a lack of technology to perform surface guidance. We have also included SGRT in our recommendation to improve accuracy in younger patients.

It is unclear why daily imaging isn't being performed to improve treatment accuracy using kV orthogonal images. However, it could be due to the technology available, please add more details about it.

We have included additional information about the usefulness of offline correction strategies. Even if daily online imaging is conducted, correction protocols like the one we implemented in our study are valuable resources that can reduce set-up time and improve patient flow.

Are patients being imaged with the MV beam, and are imaging MUs accounted for in planned doses? If images are acquired using the treatment field, why aren't daily images considered? Is the total setup error applied when images are taken?

Response: Thank you for your insightful comment. All pre-treatment imaging is conducted using an MV beam in our patient population. Our previous verification protocol, which was also implemented during this study, entailed imaging all treatment fractions. The purpose of this study, however, was to investigate whether an off-line correction strategy would be effective for this patient population. Even when online verification is performed, offline correction strategies can still provide significant benefits by reducing setup time and minimizing RTT movement to the treatment rooms for couch adjustments. These factors are especially important in busy departments where patient throughput is critical. The MU for imaging was not counted during planning.

It would be helpful to have more details about treatment planning, including the number of fields used and field arrangement.

Response: We thank the reviewer for this observation. We have included the details about the treatment planning process in the pre-treatment preparation section.

Editorial comment: Your ethics statement should only appear in the Methods section of your manuscript. If your ethics statement is written in any section besides the Methods, please move it to the Methods section and delete it from any other section. Please ensure that your ethics statement is included in your manuscript, as the ethics statement entered into the online submission form will not be published alongside your manuscript.

Response:Thank you for this comment. the Ethical statement has been moved to the Methods section.

---

## [Decision Letter · Decision Letter 1]

17 Jan 2024

Performance of an Offline Systematic Error Correction Strategy in Pediatric Patients Receiving Adjuvant Conformal Radiotherapy for Wilm’s Tumor

PONE-D-23-19186R1

Dear Dr. Mruma,

We’re pleased to inform you that your manuscript has been judged scientifically suitable for publication and will be formally accepted for publication once it meets all outstanding technical requirements.

Kind regards,

Bruno Fionda

Academic Editor

PLOS ONE

Additional Editor Comments (optional):

Reviewers' comments:

Reviewer's Responses to Questions

**Comments to the Author**

1. If the authors have adequately addressed your comments raised in a previous round of review and you feel that this manuscript is now acceptable for publication, you may indicate that here to bypass the “Comments to the Author” section, enter your conflict of interest statement in the “Confidential to Editor” section, and submit your "Accept" recommendation.

Reviewer #2: All comments have been addressed

2. Is the manuscript technically sound, and do the data support the conclusions?

Reviewer #2: Yes

3. Has the statistical analysis been performed appropriately and rigorously? 

Reviewer #2: Yes

4. Have the authors made all data underlying the findings in their manuscript fully available?

Reviewer #2: Yes

5. Is the manuscript presented in an intelligible fashion and written in standard English?

Reviewer #2: Yes

6. Review Comments to the Author

Reviewer #2: Thank you for updating the manuscript following the recommendations, I have reviewed the changes made and happy with it.

7. PLOS authors have the option to publish the peer review history of their article (what does this mean?). If published, this will include your full peer review and any attached files.

Reviewer #2: No

---

## [Editor Report · Acceptance letter]

8 Feb 2024

PONE-D-23-19186R1 

PLOS ONE

Dear Dr. Mruma, 

I'm pleased to inform you that your manuscript has been deemed suitable for publication in PLOS ONE. Congratulations! Your manuscript is now being handed over to our production team.

Kind regards, 

on behalf of

Dr. Bruno Fionda 

Academic Editor

PLOS ONE